# Anodal Transcranial Direct Current Stimulation over the Right Dorsolateral Prefrontal Cortex Boosts Decision Making and Functional Impulsivity in Female Sports Referees

**DOI:** 10.3390/life13051131

**Published:** 2023-05-05

**Authors:** Shahrouz Ghayebzadeh, Shirin Zardoshtian, Ehsan Amiri, Louis-Solal Giboin, Daniel Gomes da Silva Machado

**Affiliations:** 1Faculty of Sport Sciences, Razi University, Kermanshah 6714414971, Iraneamiri.tmu@gmail.com (E.A.); 2EFOR-CVO, 69003 Lyon, France; lsgiboin@soladis.ch; 3Research Group in Neuroscience of Human Movement (NeuroMove), Department of Physical Education, Federal University of Rio Grande do Norte, Natal 59078-970, RN, Brazil

**Keywords:** dorsolateral prefrontal cortex, sports referees, impulsivity, performance, sensitive decision making, transcranial electrical stimulation

## Abstract

We investigated the effect of anodal transcranial direct current stimulation (tDCS) over the right dorsolateral prefrontal cortex (rDLPFC) on the sensitive decision making of female team sports referees. Twenty-four female referees voluntarily participated in this randomized, double-blind, crossover, and sham-controlled study. In three different sessions, participants received either anodal (a-tDCS; anode (+) over F4, cathode (−) over the supraorbital region (SO)), cathodal (c-tDCS; −F4/+SO), or sham tDCS (sh-tDCS) in a randomized and counterbalanced order. a-tDCS and c-tDCS were applied with 2 mA for 20 min. In sh-tDCS, the current was turned off after 30 s. Before and after tDCS, participants performed the computerized Iowa Gambling Task (IGT) and Go/No Go impulsivity (IMP) tests. Only a-tDCS improved IGT and IMP scores from pre to post. The delta (Δ = post–pre) analysis showed a significantly higher ΔIGT in a-tDCS compared to c-tDCS (*p* = 0.02). The ΔIMP was also significantly higher in a-tDCS compared to sh-tDCS (*p* = 0.01). Finally, the reaction time decreased significantly more in a-tDCS (*p* = 0.02) and sh-tDCS (*p* = 0.03) than in c-tDCS. The results suggest that the a-tDCS improved factors related to sensitive decision making in female team sports referees. a-tDCS might be used as an ergogenic aid to enhance decision performance in female team sports referees.

## 1. Introduction

Transcranial direct current stimulation (tDCS), also known as a neurodoping or neuroenhancement technique, has been widely used in sports-related studies over the last two decades [1,2]. tDCS induces its effects by changing the excitability of the target areas in the brain in a polarity-specific manner [3]. It has been shown that anodal and cathodal stimulation increase and decrease the excitability of the areas of interest, respectively [4]. The positive effects of tDCS on muscular strength and whole-body endurance performance, power output, and cognitive function in healthy and clinical populations have been reported in several studies [5,6,7,8,9,10,11]. These promising results have paved the path for the investigation of the efficacy of tDCS in other target populations and conditions related to sports activities.

One such target population is sports referees, who are an integral part of any sports event, since their decisions might have a big impact on the results of a game [12]. For many referees, sports events bring tempting moments, high expectations for making the right decisions, and ample potential to influence the outcome [12,13,14]. Such circumstances particularly exist for referees who need to consider multiple hints as they interact with players, coaches, and even spectators during sports events [12]. Similarly, decisions made by referees at major championship events may also be regarded as sensitive, since they are at the center of attention of many individuals nationally and internationally [13,14]. Accordingly, referees’ decisions could be considered a type of sensitive decision making [12]. Neuroscience research has identified two types of sensitive decisions that are known to be coded differently by the human brain [15]. The first type involves analytical and logical decisions, which are coded and saved by the dorsolateral prefrontal cortex (DLPFC). The second type is a decision under uncertainty, which requires visual cues and is coded in the orbitofrontal cortex [16,17]. These decisions often must be made quickly due to certain internal and/or external pressures. During a given sports event, major internal factors such as bias, stress, emotion, impulsivity, and reaction time can influence a referee’s decisions [14]. Among them, impulsivity and reaction time are known to be significant factors affecting referees’ decisions [18].

Classically, impulsivity was defined as the tendency to respond to a given stimulus with a low or inadequate degree of pondering, precaution, or control [19]. This definition implies that impulsivity is a dysfunctional state leading to unfavorable outcomes [20]. However, it has been shown that impulsive behavior does not always result in negative consequences; in contrast, under certain circumstances, it may lead to favorable responses [19,20]. Accordingly, the term “functional impulsivity” was coined to describe this aspect of impulsivity and its relevance to the decision-making process [19]. Interestingly, functional impulsivity implies that although deliberate, careful, and forethought responses could be considered “safe and sure”, under many real-world conditions, it can culminate in unfavorable consequences and missed opportunities [19,20,21]. For instance, we often make quick driving reactions when other drivers behave carelessly. These certainly prevent disasters from happening, but we do them without much prior thinking. Similarly, in complex situations during sports events, delayed reactions and decisions can cause undesirable responses from the stakeholders. This is exactly when the referees’ impulsivity can be effective and spare them from hostile reactions expressed by others [20,22]. Furthermore, a referee’s decision must be made quickly and compatibly with the actions witnessed during sports events. Otherwise, the decision may lead to protests from athletes, trainers, other referees, and spectators [12,23]. Hence, functional impulsivity could also be pertinent to this aspect of the decision making in sports referees in which they may tend to respond quickly, leading to optimal outcomes but causing no unfavorable consequences [19].

Astonishingly, recent findings have demonstrated that tDCS could have a modulatory effect on decision making and impulsivity in samples of healthy adults and individuals with gambling disorders [24,25,26,27]. Among different brain target areas, the prefrontal cortex (PFC) and DLPFC have been deemed to play a pivotal role in risky decision making; this conjecture seems to be more conceivable for the right hemisphere of the brain [24,27]. In this regard, Soyata et al. [26] reported that applying 20 min of anodal tDCS over the right DLPFC area improved the decision making and cognitive flexibility of male participants with gambling disorders. In another study, Ota et al. [28] showed that anodal stimulation of the right DLPFC resulted in a more conservative strategy to avoid the risk of no reward in healthy adult males. Similarly, in a systematic review of the literature, Khaleghi et al. [25] confirmed the efficacy of the neuroregulation of the DLPFC area on risk-taking behavior in healthy individuals.

Despite these promising effects and advances in the field, there are still some gaps that need to be filled in this context. For example, considering the importance and sensitivity of sports referees’ decisions, no previous study has tested the efficacy of neuromodulatory interventions such as tDCS in factors affecting the decisions of sports referees. Needless to say, the nature of the sports events and the atmosphere in which the referees make their decisions are not comparable with other situations, which makes it difficult to generalize the results of previous studies to sports referees. In this context, it has been shown that individual differences, task difficulties, and external cues might affect the effectiveness of tDCS in modulating the variables related to decision making [29,30,31]. In addition, it has been demonstrated that there is a sex difference in the execution of responses after DLPFC stimulation by the use of tDCS, which indicates that caution must be exercised when generalizing the research outcomes to both sexes. It is worth noting that most of recent studies in this field have been conducted on male participants, showing the necessity of conducting research studies on female participants [32,33,34]. This is important, considering the differences between sexes, with implications for neuromodulation [34], which results in differences between men and women in tDCS-induced electric field, on the target area; for different montages and current intensities [33], changes in corticospinal excitability [35], functional connectivity [36], and clinical findings [32,37].

Therefore, based on the dearth of information regarding the effectiveness of tDCS in sports referees’ decision-making performance—in particular female sports referees—and sex-related implications for neuromodulation techniques, the aim of this study was to investigate the effect of tDCS applied over the right DLPFC area on the decision making, impulsivity, and reaction time of female team sports referees. We hypothesized that anodal stimulation of the right DLPFC would improve sensitive decision making, increase functional impulsivity, and shorten the reaction time of female sports referees.

## 2. Materials and Methods

### 2.1. General Experimental Design

The study was randomized, counterbalanced, double-blind, and sham-controlled with a within-subject design in which each participant visited the laboratory on 4 days interspersed by a 72-h interval. During the first visit, participants were familiarized with the study purpose and procedures, screened for eligibility, signed the written consent form, and were familiarized with the tDCS intervention and measurement of the study variables. In the next three experimental sessions, participants were randomly assigned to one of the three experimental conditions: (1) anodal tDCS (a-tDCS), (2) cathodal tDCS (c-tDCS), or (3) sham tDCS (s-tDCS). In each experimental session, participants first performed the Iowa Gambling Task (IGT), and Go/No Go Impulsivity tests on a computer under identical conditions. Then, tDCS was applied for 20 min (see tDCS section for details). Immediately after tDCS, the IGT and Go/No Go Impulsivity tests were repeated. Neither the participant nor the researcher applying the outcome measures assessment was aware of the tDCS condition. The study design is presented in Figure 1.

### 2.2. Participants

A total of 24 volunteer female team sports referees from various team sports, such as soccer, futsal, volleyball, basketball, and handball, aged 18–38 years old (mean: 28 ± 3.25 years of age) participated in this study. The inclusion criteria were (a) being right-handed, (b) having no history of clinical impairments or neurological disorders, (c) not currently using an external or internal electrical stimulator in the body, and (d) having at least 3 years of experience with national and international sports competitions. The sample size calculation was performed a *priori* using G*Power software, version 3.1.9.2 (Universität Kiel, Kiel, Germany) [38]. The sample size was calculated using the following command: test family: F tests; statistical test: ANOVA: repeated measures, within factors; α error probability: 0.05; power (1-β err prob): 0.80; effect size f: 0.3, number of groups = 1, number of measurements = 3. The statistically required sample size was 20, but considering a possible sample loss of 20%, a total of 24 participants were recruited. The experimental procedures were reviewed and approved by the Institutional Ethics Committee and conducted following the Helsinki Declaration on human research ethics. We decided to include only women in order to have a more homogeneous sample, as there are differences between sexes in terms of neuromodulation [34], which impact the tDCS-induced electric field on the target area for different montages and current intensities [33], resulting in different changes in corticospinal excitability [35], functional connectivity [36], and clinical findings [32,37] between men and women.

### 2.3. Randomization, Concealment, and Blinding Process

In this study, the order of participants’ exposure to 3 different conditions was randomized by the Latin squares’ method. To do so, first, using the site www.random.org, a number between 1 and 24 was randomly allocated to each participant as an identification code. Then, the English letters A, B, and C were assigned to the three intervention conditions, and a Latin square was created. In this case, a Latin square with three rows and three columns was created. Finally, participants 1 to 8 were placed in the sequence of the first row, participants 9 to 16 were placed in the sequence of the second row, and participants 17 to 24 were placed in the sequence of the third row. Moreover, the investigator and participants were blinded to the types of tDCS used in each session. To do so, an individual outside the research team was responsible for applying electrical stimulation in three experimental sessions. To blind the participants, after they sat on a custom-made chair, the stimulator was hidden from their view and wrapped by a cover, and the electrodes were placed on the target areas by the researcher applying tDCS. To blind the investigator, in each experimental session, immediately before the beginning of the tDCS intervention, the investigator left the laboratory and returned to the test site after the termination of the stimulation time, removal of the electrodes, and turning off the stimulator.

### 2.4. Iowa Gambling Task (IGT)

We used the Persian adaptation of the computerized IGT (Medina Tebgostar; Tehran, Iran), in which each participant was given a virtual loan of $2000 and was expected to win as much as possible during the subsequent 100 trials. The gambling task was presented with four sets of card decks identified as A–D. Choosing cards from the A and B decks might lead to greater gains or losses (>$100) as opposed to pulling cards from the C and D decks. After all gambling tasks, the software summed up the scores and registered the total gains, losses, total score, and the frequency of selecting cards from the A–D decks for each participant [39,40].

### 2.5. Impulsivity (IMP) and Reaction Time (RT)

The digital impulsivity games (Medina Tebgostar; Tehran, Iran) involved 100 efforts chosen at the discretion of the participant as “*Go*” or “*No Go*” [41]. During each of the subsequent 100 games, at some points, the monitor displayed either the letter “*P*” or “*R*” in a square. The participant was required to click on “*P*” as soon as possible to get credit for a correct choice but to do nothing if the letter “*R*” was displayed. The monitor also displayed an “*R*” if the participant made a wrong choice. After the games, the software displayed the scores for total impulsivity, reaction time, and the best reaction time for any of the games played by each participant [42]. A higher score in the Go/No Go test indicated better functional impulsivity. The impulsivity score and the best reaction time of the participants were used for statistical analysis.

### 2.6. Transcranial Direct Current Stimulation (tDCS)

We applied tDCS with 2 mA targeting the right DLPFC with either anodal (a-tDCS) or cathodal (c-tDCS) stimulation. We used an automatic battery-driven stimulator (Neurostim, Medina Tebgostar, Tehran, Iran) with two 5 × 7 cm carbon electrodes (electrode area = 35 cm^2^; current density = 0.057 mA/cm^2^) covered by saline-soaked surface sponges to deliver the tDCS over the target area of the scalp, held in place with elastic strips. The electrode positions were based on the EEG 10/20 system. For a-tDCS, the anodal electrode was placed over F4, corresponding to the right DLPFC, and the cathodal electrode was placed over the right supraorbital area (Fp1). For c-tDCS, the cathodal and anodal electrodes were placed over F4 and Fp1, respectively [43]. For active stimulations, a 2-mA direct current was applied for 20 min with a 30 s ramp-up and ramp-down of the electric current at the beginning and end of stimulation, respectively. For sham tDCS (s-tDCS), the electrodes were positioned at the same scalp locations as a-tDCS, following identical procedures, but the current was turned off after 30 s. The sham protocol has been demonstrated to be adequate for blinding, as it induces identical scalp sensations as the active tDCS [26,44]. To evaluate the effectiveness of the study’s blinding process, we administered a researcher-made questionnaire to the participants on each of the three days of the experiments, asking the participants’ perception of the stimulation mode they received during each of the three tDCS sessions. They checked their responses on the questionnaire as *anodal*, *cathodal*, or *sham*. This was used to calculate the ‘correct guess rate’ and active stimulation guess rate. According to recent literature, the correct guess rate at the end of the study indicates the percentage of participants that successfully guessed their experimental condition, which might lead to a misleading interpretation of blinding effectiveness [45,46]. It has been suggested to report the “active stimulation guess rate”, which indicates the percentage of participants who guessed that they received the active treatment [46]. Therefore, although we report both correct and active stimulation guess rates, we will consider the latter as the measure of blinding effectiveness [46].

#### tDCS Modeling

The brain current flow during tDCS was calculated using a finite element model (FEM) following the standard pipeline in SimNIBS 4.0.0 [47]. The magnetic resonance imaging (MRI) MNI 152 head model available in the software was used. MRI data were segmented into surfaces corresponding to the white matter (WM), gray matter (GM), cerebrospinal fluid (CSF), skull, and skin. The electrical conductivities of each segment were determined according to values previously established as follows: WM = 0.126 Siemens/meter (S/m), GM = 0.275 S/m, CSF = 1.654 S/m, bone = 0.010 S/m, and skin/scalp = 0.465 S/m [48]; rubber electrode = 29.4 S/m; and saline-soaked sponges = 1.000 S/m. All information concerning the respective tDCS montages was entered into the software as follows: current intensity = 2 mA; electrode position (+F4/−AF3 and −F4/+AF3); electrode and sponge sizes (5 × 7 cm); electrode thickness = 1 mm; sponge thickness = 5 mm. The results of the simulations are presented in Figure 2 in terms of the electric field strength and radial electric field (normal to the cortical surface), both of which are most important for neuromodulatory effects [49]. As shown in Figure 2, the study montage reached our target areas with enough electric current strength to induce a neuromodulatory effect. Furthermore, the target areas were stimulated with the desired polarity (i.e., anodal current) to induce excitatory effects in the target regions. Other areas such as the ventromedial prefrontal cortex and frontopolar area were also stimulated in the current path from the anodal to the cathodal electrode.

### 2.7. Statistical Analyses

The normal distribution of each dataset was evaluated by the Shapiro–Wilk normality test. All values in the figures are presented as the means ± standard deviations (M ± SD). A two-way repeated-measures ANOVA was performed (3 × 2 factorial design; 3 stimulation conditions and 2 time points) was performed on the mean value of the IGT, IMP, and RT. We also further explored the data by calculating delta (Δ) values (post–pre) for each outcome variable and compared them using a repeated-measures ANOVA [50]. All ANOVAs were followed by a post hoc test for pairwise comparisons using Bonferroni correction for multiple comparisons. In case of a violation of the assumption of sphericity, Greenhouse–Geisser epsilon correction was applied. Partial eta squared (η^2^_p_) was used as a measure of the effect size for the ANOVAs and interpreted as small (0.01–0.059), medium (0.06 to 0.139), or large (≥0.14). Cohen’s d calculation of the effect size was also used for pairwise comparison and interpreted as small (0.20–0.49), medium (0.50–0.79), or large (≥0.80). The statistical analyses were performed using Statistica 8.0 (StatSoft, Tulsa, OK, USA), and *p* ˂ 0.05 was adopted.

## 3. Results

The overall results of the IGT, IMP, and RT at pre- and post tDCS intervention are presented in Table 1 and Figure 3. More than 87% of the end of study guesses were incorrect, which suggests our blinding protocol was effective.

### 3.1. Effect of tDCS on Iowa Gambling Task performance

For the IGT performance (Figure 3A, Table 1), there was a trend towards significance for the main effect of time (F_(1,23)_ = 4.06, *p* = 0.056, η^2^_p_ = 0.15) and a significant ‘condition × time’ interaction (F_(2,46)_ = 4.33, *p* = 0.019, η^2^_p_ = 0.158), with no significant main effects of condition (F_(2,46)_ = 0.44, *p* = 0.65, η^2^_p_ = 0.019). The Bonferroni post hoc test showed a significant increase from pre- to post intervention only in a-tDCS (*p* = 0.013, d = 0.34). The post-intervention value for IGT was higher in a-tDCS (*p* = 0.016, d = 0.26) and sh-tDCS (*p* = 0.037, d = 0.26) compared to c-tDCS, with no difference between a-tDCS and sh-tDCS (*p* = 1.00). Finally, there was no difference in baseline performance among conditions (all ps > 0.12).

There was a significant difference among the conditions for the ΔIGT value (F_(2,46)_ = 4.33, *p* = 0.019, η^2^_p_ = 0.158). The Bonferroni post hoc test revealed that the ΔIGT was significantly higher in the a-tDCS condition than in the c-tDCS (*p* = 0.022, d = 0.66) condition. No significant difference was found between sh-tDCS and a-tDCS (*p* = 0.11) or c-tDCS conditions (*p* = 1.00) (Figure 3B, Table 1).

### 3.2. Effect of tDCS on Impulsivity (IMP)

For IMP performance, there was a trend towards significance for the main effect of condition (F_(1.5,36.5)_ = 3.14, *p* = 0.052, η^2^_p_ = 0.120) and a significant main effect of time (F_(1,23)_ = 26.5, *p* < 0.001, η^2^_p_ = 0.536) and ‘condition × time’ interaction (F_(2,46)_ = 5.21, *p* = 0.009, η^2^_p_ = 0.185) (Figure 3C, Table 1). The Bonferroni post hoc test showed that IMP at baseline was similar between the a-tDCS and c-tDCS conditions (*p* = 1.00), but both a-tDCS (*p* = 0.005, d = 0.36), and that in c-tDCS (*p* = 0.044, d = 0.29) was higher than in sh-tDCS. IMP increased significantly from pre to post only in a-tDCS (*p* < 0.001, d = 0.55) but not in c-tDCS (*p* = 0.52) or sh-tDCS (*p* = 1.00). In addition, IMP values after a-tDCS (*p* < 0.001, 0.78) and c-tDCS (*p* = 0.002, d = 0.44) were higher than after sh-tDCS.

There was a significant difference among the condition ΔIMP values (F_(2,46)_ = 5.21, *p* = 0.009, η^2^_p_ = 0.185). The Bonferroni post hoc test showed that the ΔIMP was significantly higher in the a-tDCS condition than the sh-tDCS condition (*p* = 0.01, d = 0.91) and trended towards significance compared to c-tDCS (*p* = 0.068, d = 0.65). No significant differences were observed between other conditions (*p* > 0.05) (Figure 3D, Table 1).

### 3.3. Effect of tDCS on Reaction Time (RT)

Finally, for RT (Figure 3E, Table 1), there was significant main effect of time (F_(1,23)_ = 7.62, *p* = 0.011, η^2^_p_ = 0.249) and ‘condition × time’ interaction (F_(2,46)_ = 5.08, *p* = 0.010, η^2^_p_ = 0.181) but no significant main effect of condition (F_(2,46)_ = 0.155, *p* = 0.86, η^2^_p_ = 0.007). The Bonferroni post hoc test showed no significant difference between any time points or conditions.

There was a significant difference among the conditions for ΔRT (F_(2,46)_ = 5.08, *p* = 0.010, η^2^_p_ = 0.181). The Bonferroni post hoc test showed that the ΔRT values in a-tDCS (*p* = 0.018, d = 0.85) and sh-tDCS (*p* = 0.03) were significantly higher than in c-tDCS, while we found no significant differences in ΔRT values between a-tDCS and sh-tDCS (*p* = 1.00) (Figure 3F, Table 1). 

## 4. Discussion

In this randomized, double-blind, and sham-controlled study, we investigated the effect of tDCS on sensitive decision making, impulsivity, and reaction times in female sports referees. To the best of the authors’ knowledge, this was the first study to use tDCS with the aim of improving sensitive decision making in sports referees. We obtained promising results regarding the efficacy of tDCS. As shown in Figure 3, the results indicate that anodal stimulation over the right DLPFC area significantly improved the participants’ decision making, as evidenced by the increased delta IGT net score compared to that for cathodal tDCS. It has recently been shown that the interaction between the limbic and cognitive loops is the primary neural process underlying decision making [18]. In this model, DLPFC has a pivotal role in the cognitive loop, controlling executive and motor functions [18,51]. It appears that in our study, modulating the activity of the right DLPFC by anodal stimulation boosted the participants’ decision making. This finding is supported by published evidence indicating that decision-making processes are encoded in two separate neural circuits in the brain, in which the DLPFC predominantly affects highly sensitive decisions [15]. This is interesting because, as mentioned in the Introduction, individual differences and task difficulty might have a modulatory effect on response to tDCS intervention and should therefore be carefully considered when interpreting the results [52,53]. In our study, the referees were classified as *interactor* referees according to the recent classification provided by Kittel et al. [12]. *Interactor* referees have much higher information processing and cognitive demands to decide in comparison to *monitor* and *reactor* referees [12]. In addition, it has been conjectured that there are many unwritten rules that *interactor* referees consider when making decisions, indicating the difficulty of the decision-making process for them [12]. Taken together, it seems that tDCS intervention in these specific sports referees, who are considered to be high performers in their expertise, could positively modulate the cognitive and information-processing demands associated with making decisions and improve the performance of the referees in a related task.

Our results also indicate improvement in the participants’ impulsivity under anodal stimulation of the right DLPFC compared to the sham mode as measured by the net scores for each participant on the Go/No Go test (Figure 3). Impulsivity refers to an individual’s tendency to do perform tasks relatively quickly and without prior preparation or planning [54,55]. Recently, it has been established that the right DLPFC plays a critical role in the inhibitory control of decision-making processes [56,57,58]. Two different mechanisms have been proposed for this inhibitory control of the decision-making process. First, the anatomic connection and coactivation between DLPFC and the orbitofrontal cortex provide a window to think about the functional interaction between the two regions as one of the mediators of decision making under uncertainty [56,57,58]. Second, based on the association between the subjective experience of time and impulsivity and the indispensable role of DLPFC in time processing, virtual lesion-like effects of DLPFC have been proposed as an alternative mechanism [56,59,60]. However, in this study, it is not yet clear which of these mechanisms contributed the most to the impulsivity role of sports referees. Thus, further in-depth research is warranted to gain better insight into the mechanisms involved in modulating the DLPFC aided by tDCS. From another perspective, as previously mentioned, impulsivity is classified as functional and dysfunctional, and it is now clear that these two constructs have different cognitive mechanisms and are not correlated across individuals [19]. Tt seems that the characteristics of the referees as being *interactors*, who are deemed to be high performers, play a role in the impulsive response to the anodal tDCS intervention applied in this study. The increment in the IMP in our study was accompanied by an increase in the IGT under anodal stimulation, which might be interpreted as tDCS-induced functional impulsivity leading to better performance in a decision-making task in referees. This puts more emphasis on the role of individual expertise and its interaction with different tDCS paradigms.

The data for the participants’ reaction times demonstrates significantly longer delta RT under the cathodal stimulation than the other tDCS modes (Figure 3). In this context, most previous studies have reported inhibitory effects of cathodal stimulation. This fact implies that cathodal stimulation may inhibit a specific neural transmission, thereby prolonging a person’s reaction time [61]. Numerous variables, including the nature and magnitude of stimulation, duration and frequency of experiment, neural adaptation, and central information processing, may influence an individual’s reaction time [62]. Each of these variables can potentially shorten or prolong an individual’s reaction time after a given stimulus. Given the above facts, variations in reaction times may be indicative of different information-processing events in the brain. Interestingly, our results raise an argument about whether tDCS-induced prolongation in reaction time, which could somehow be considered reactive response inhibition, is favorable for the decision-making process in interactor referees. In this context, it has been demonstrated that the ability to inhibit an already initiated response is essential to make deliberate decisions, in particular when there is an emergent process requiring a balance of myriad variables such as what we see in decisions made by *interactor* referees during a match [63]. According to our results, we are not able to draw a conclusive interpretation regarding the abovementioned question, since in our study, delta RT was longer under cathodal stimulation not accompanied by an improvement in IGT and IMP in sports referees. Therefore, more research is required to determine whether there are synergistic effects of IGT and IMP increment and RT prolongation on performance-related variables in *interactor* referees.

The limitations of the present study include the fact that we did not use any measure of brain activity that could help to elucidate possible mechanisms related to tDCS-induced changes, such as electroencephalography (EEG), functional near-infrared spectroscopy (fNIRS), etc. Furthermore, since this study was conducted in female referees and the tDCS-induced electrical field and its related changes in neuronal excitability and activity may vary between sexes, the generalizability of the present results for male sports referees remains to be tested. Future studies should also test the effect of repeated sessions of tDCS, as such a condition might induce greater long-term potential-like effects. Finally, although we used validated assessment tools, they present low ecological validity. Hence, future studies should use more ecological tasks related to the activities of team sports referees (e.g., involving decisions related to specific sportive contexts).

## 5. Conclusions

This study showed, for the first time, that targeting the right DLPFC with anodal tDCS might improve the sensitive decision-making process in female sports referees. Such results were not obtained from the cathodal and sham modes of tDCS. tDCS may improve the central information processing involved in decision making and impulsivity, which warrants further investigation to better understand its mechanisms. The tDCS technique could potentially pave the way for considering non-invasive brain stimulation as a new strategy to enhance the performance of referees in sensitive decision making. Considering that the novel approach is in its early stages, there is no consensus yet on its efficacy and generalizability to the real world of sports. Thus, from a practical standpoint, further studies with rigorous designs are warranted to corroborate our findings on the application and usefulness of tDCS in sports events. Moreover, the effect of tDCS could also be tested in athletes in sports in which decision making plays a critical role in the sporting outcome. 

## Figures and Tables

**Figure 1 life-13-01131-f001:**
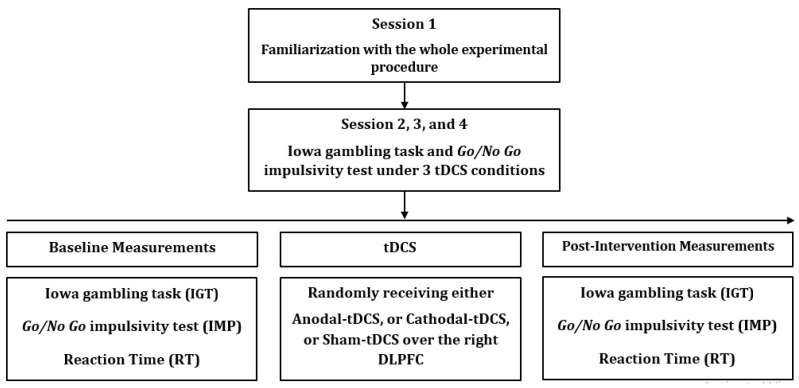
Study flow chart. tDCS = transcranial direct current stimulation; DLPFC = dorsolateral prefrontal cortex.

**Figure 2 life-13-01131-f002:**
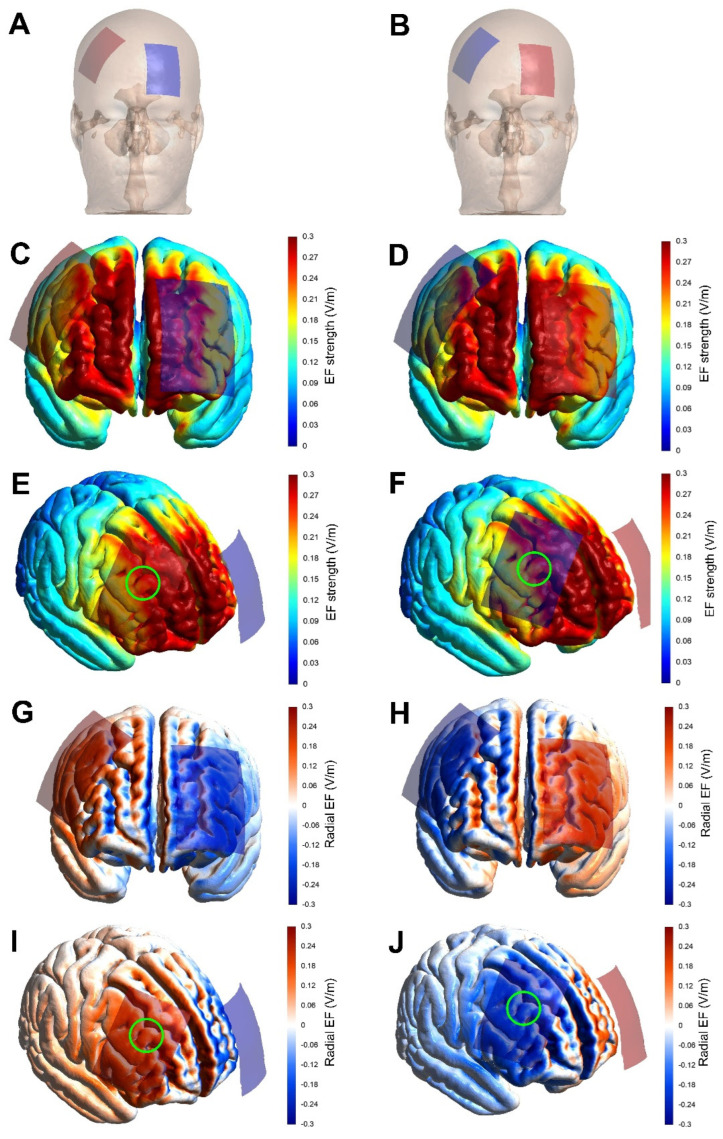
Strength and radial component of the electric field induced by tDCS. Anodal (red rectangle) and cathodal (blue rectangle) electrodes (5 × 7 cm) placed over the scalp (**A**,**B**). Finite element models derived from magnetic resonance imaging in a head model (MNI152) of the strength and radial (normal to the cortical surface) component of the electric field induced by tDCS. Electrode montage targeting stimulation with anodal and cathodal tDCS of the right dorsolateral prefrontal cortex (**left** and **right** columns, respectively). Electric field strength is presented in the color-coded figures (**C**–**F**), with hotter colors indicating stronger electric field and colder colors indicating weaker electric field. The radial electric field is presented in the color-coded figures (**G**–**J**), where red color represents the electric current flowing into the cortex (i.e., inducing excitatory effects), and blue color represents the electric current flowing out of the cortex (i.e., inducing inhibitory effects). The study montage reached the target areas with enough electric current strength to induce a neuromodulatory effect, as shown in figures electric field (**E**,**F**; green circles roughly representing the target areas). Furthermore, the target areas were stimulated with the desired polarity (i.e., anodal current) to induce excitatory effects in the target regions, as shown in panels (**I**,**J**) (green circles roughly representing the target areas).

**Figure 3 life-13-01131-f003:**
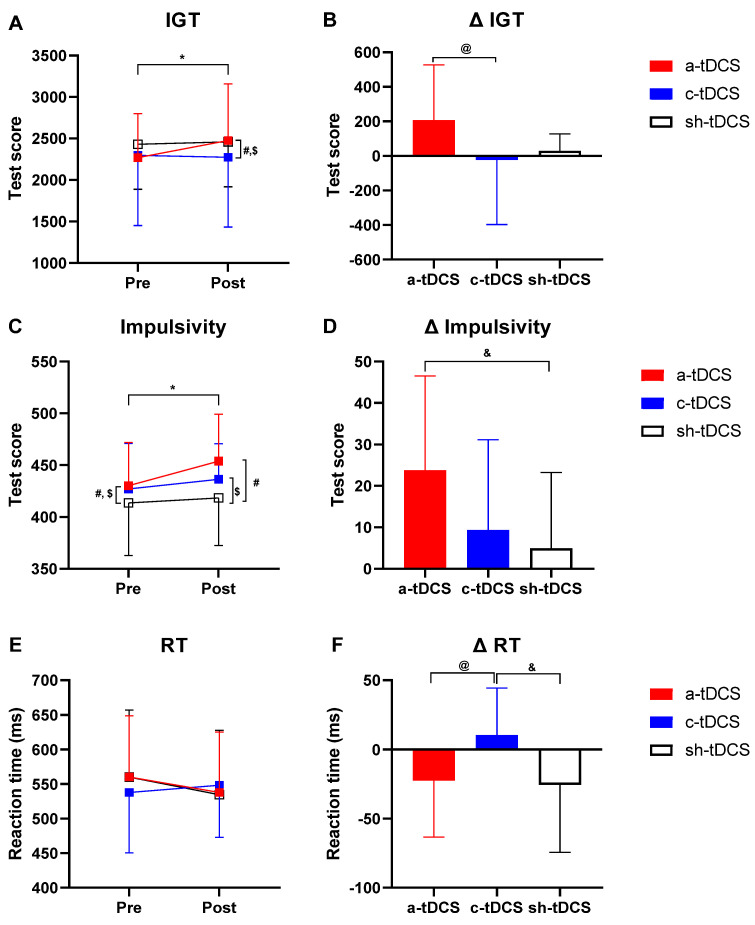
Effect of transcranial direct current stimulation (tDCS) on decision making, impulsivity, and reaction time. The figure depicts absolute and delta scores for the Iowa Gambling Task (IGT; (**A**,**B**), respectively), impulsivity ((**C**,**D**), respectively), and reaction time ((**E**,**F**), respectively) before and after anodal (a-tDCS), cathodal (c-tDCS), or sham tDCS (sh-tDCS). * = significant difference from pre- to post anodal tDCS (*p* ≤ 0.01); ^#^ = significantly different from anodal tDCS at the same time point (*p* ≤ 0.02); ^$^ = significantly different from cathodal tDCS at the same time point (*p* < 0.05); ^@^ = significantly different from anodal tDCS (*p* ≤ 0.02); ^&^ = significantly different from sham tDCS (*p* ≤ 0.03).

**Table 1 life-13-01131-t001:** Effect of transcranial direct current electrical stimulation (tDCS) on decision making, impulsivity, and reaction time in female sports referees (n = 24).

Variable	Time	tDCS Condition		
		Anodal	Cathodal	Sham
Iowa Gambling Task ($)	Pre	2269.0 ± 529.9	2296.9 ± 846.2	2430.2 ± 542.1
	Post	2476.3 ± 680.4 *^,#,$^	2273.8 ± 841.0	2459.6 ± 541.8
	Delta	207.3 ± 319.3 ^@^	−23.1 ± 374.4	29.4 ± 98.0
Impulsivity (Scores)	Pre	430.2 ± 41.7 ^#,$^	427.1 ± 43.9	413.5 ± 50.6
	Post	454.0 ± 45.2 *^,#,$^	436.5 ± 34.3	418.5 ± 46.1
	Delta	23.8 ± 22.8 ^&^	9.4 ± 21.8	4.9 ± 18.3
Reaction time (ms)	Pre	560.0 ± 97.2	537.9 ± 87.6	560.4 ± 88.4
	Post	534.6 ± 93.4	548.3 ± 75.3	537.9 ± 87.5
	Delta	−25.4 ± 49.0 ^@^	10.4 ± 33.9 ^&^	−22.5 ± 40.8

Note: data presented as mean ± standard deviation; * = significant difference from pre- to post anodal tDCS (*p* ≤ 0.01); ^#^ = significantly different from anodal tDCS at the same time point (*p* ≤ 0.02); ^$^ = significantly different from cathodal tDCS at the same time point (*p* < 0.05); ^@^ = significantly different from anodal tDCS (*p* ≤ 0.02); ^&^ = significantly different from sham tDCS (*p* ≤ 0.03).

## Data Availability

The data generated and/or analyzed during the current study are available from the corresponding author upon reasonable request.

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
