# Peer review of "Anodal Transcranial Direct Current Stimulation over the Right Dorsolateral Prefrontal Cortex Boosts Decision Making and Functional Impulsivity in Female Sports Referees"

_life, 2023, doi:10.3390/life13051131_

Round 1

Reviewer 1 Report

Well written presentation of the study conducted by the authors. Detailed methodology section and good discussion with clearly state of the study limitations. I feel that the manuscript could be publihsed in its present form.

Author Response

The detailed response is presented in the file attached. Thank you.

Reviewer 2 Report

To begin with, I would like to express my gratitude for the chance to review the research article titled "Anodal tDCS Over the Right DLPFC Boosts Decision-Making and Functional Impulsivity in Female Sports Referees." It is an honor for me to be chosen as a contributor to the peer-review process for the journal "Life."

I appreciate the trust placed in me to undertake this essential task and am eager to provide a comprehensive and constructive review.

In this study examined the effect of anodal transcranial direct current stimulation (tDCS) over the right dorsolateral prefrontal cortex (rDLPFC) on the decision-making of female team sports referees. 24 referees participated in a randomized, double-blind, crossover, and sham-controlled study where they received either anodal, cathodal, or sham tDCS in different sessions. The computerized Iowa gambling task and Go/No Go impulsivity tests were performed before and after tDCS. The results showed that only anodal tDCS improved scores related to decision-making and reaction time in female team sports referees. The study suggests that anodal tDCS could be used as an aid for enhancing the decision-making performance in female team sports referees.

INTRODUCTION

The introduction of the study provides a general overview of the research topic and background information on transcranial direct current stimulation (tDCS) and its potential effects on decision-making and impulsivity. However, it could be improved by providing more specific details about the research gap or problem that this study aims to address. For example, the introduction could have included a more detailed discussion of the limitations of previous studies on tDCS and decision-making in sports referees, which would help to justify the need for this study. Additionally, the introduction could have provided a clearer statement of the research question or hypothesis that this study aims to answer. A well-defined research question or hypothesis would help to guide the reader's understanding of the study's purpose and significance.

DESIGN

The study design used in this research is generally rigorous and well-controlled. However, there are some potential areas for improvement. For example;

The sample size of the study was relatively small, with only 24 female sports referees participating. A larger sample size would increase the statistical power of the study and improve the generalizability of the findings. 

Additionally, the study only included female sports referees, so it is unclear whether these findings would generalize to male referees or other populations. Future studies could include a more diverse sample to test for potential gender or population differences in response to tDCS.

RESULTS

The presentation of results in this study is generally clear and well-organized. However, there are some potential areas for improvement. For example, the results section could have included more detailed descriptions of the statistical analyses used to test the study hypotheses. Providing more information about the statistical tests would help readers to better understand the strength and significance of the study findings. 

Additionally, while the study includes several figures and tables to present the data, some of these figures could be improved by providing clearer labels or annotations. 

Finally, while the authors do a good job of summarizing their main findings in the results section, they could have provided a more detailed discussion of how these findings relate to previous research on tDCS and decision-making in sports referees. A more thorough discussion would help readers to better understand the implications and significance of the study's findings.

MORE ECOLOGICAL TASK SUGGESTION.

I would like to make a reflection that serves as a suggestion for future studies. The tasks used to measure decision-making and impulsivity are tasks implemented in the laboratory and are not very ecological. As the authors indicate, making decisions in a stadium with a multitude of people is very different from doing so in the laboratory. Therefore, I propose that the authors design a contextualized task where the referee is placed in the context where the decision has to be made by adding a context to the decisions. For example:

Context: "You are refereeing the championship final. Your family and friends have come to see you. There is a doubtful foul against the favorite team. The stadium begins to shout loudly.

Plausible decisions are shown:

  1. I indicate that there is an infringement.

  2. I let the game continue.

  3. I consult with my fellow referees.

Consequences (in favor or against the favorite team's supporters).

  • In favor: The fans applaud your decision and are happy.

  • Against: The fans begin to insult you and are angry.

I recommend reviewing this article on risk decision-making during adolescence by Dr. Rodrigo. Rodrigo, M. J., Padrón, I., De Vega, M., & Ferstl, E. C. (2014). Adolescents’ risky decision-making activates neural networks related to social cognition and cognitive control processes. Frontiers in human neuroscience, 8, 60. https://www.frontiersin.org/articles/10.3389/fnhum.2014.00060/full

I have thoroughly enjoyed reviewing your work and I am certain that your paper will significantly contribute to the field with the proposed revisions. I wish you the very best in your ongoing research endeavors and eagerly anticipate the publication of your upcoming works.

Best regards,

Author Response

(The authors gave the same response as above.)

Reviewer 3 Report

Dear Authors.

The following is a review of the article entitled “Anodal tDCS Over the Right DLPFC Boosts Decision-Making and Functional Impulsivity in Female Sports Referees”. Thank you very much for thinking of me as a reviewer for this study.

After carefully reading the manuscript, I set forth comments and suggestions for the authors:

Their work is very interesting, and I congratulate the authors on a unique study.

The manuscript is excellent. Good summary and accurate key words. The introduction is correct, as is the methodology used. The exposition of the results is good, and the discussion is excellent.

It is recommended to eliminate the acronyms in the title and to extend the practical application to other areas of sport.

Congratulations for your study.

Best regards

Minor editing of English language required

Author Response

(The authors gave the same response as above.)
